# Learning AND–OR Templates for Compositional Representation in Art and Design

Liaoruxing Zhang[1], Xin Jin[1,2,*], Wenbo Yuan[1], Chenyu Fan[1] and Song-Chun Zhu[2,3,4]

[1]Beijing Electronic Science and Technology Institute, Beijing, China
[2]State Key Laboratory of General Artificial Intelligence, Beijing Institute for General Artificial Intelligence (BIGAI), Beijing, China
[3]School of Intelligence Science and Technology, Peking University, Beijing, China
[4]Department of Automation, Tsinghua University, Beijing, China

## Abstract

This work proposes a compositional AND–OR template for art and design that encodes the part–relation–geometry organization of images in a structured and interpretable form. Within a maximum-entropy log-linear model, we define a unified consistency score as log-likelihood gain against a reference distribution and decompose it into term-level evidence, enabling an evidence-to-prescription mapping for actionable composition guidance. Learning is performed by a penalized EM-style block-pursuit with sparsity and local mutual exclusivity: object templates are learned first and reused as scene terminals to induce scene templates. A semi-supervised structural expansion, which is triggered by matching gain and structural-consistency thresholds, bootstraps new branches from unlabeled, high-quality images. Evaluations on a curated compositional dataset and AVA/AADB themes show strong agreement with expert paradigms, interpretable parse trees, and competitive performance with deep baselines while exhibiting higher alignment with human ratings. The learned templates also act as lightweight structural conditions to steer AIGC generation and layout design. Overall, the framework delivers a transferable structural prior with favorable data/parameter efficiency and a unified pathway for explainable visual assessment and generation.

## 1 Introduction

Art and design are markedly structured. Existing tools for visual assessment and creative assistance often rely on large-scale annotations and black-box representations, making it hard to turn the question of why something is good, and how to improve it, into a traceable, actionable chain of evidence (Murray et al., 2012; Simonyan & Zisserman, 2014; He et al., 2016; Dosovitskiy et al., 2020). They also struggle with cross-scene transfer and few-shot generalization. We advocate a recomposable structured representation: reusable parts and constraints that capture how visual elements are organized in art and design, so a model can learn "general structure" while producing executable, task-specific recommendations.

We propose a compositional representation based on AND–OR templates for art and design (Si & Zhu, 2013; Zhu et al., 2007). AND nodes encode relational and geometric constraints that must be satisfied jointly; OR nodes represent alternative structural variants; terminals are observable primitives in art and design with statistical attributes. Under a maximum-entropy log-linear framework, we define image–template consistency as the log-likelihood gain relative to a reference distribution, yielding a unified scorer. The scorer decomposes term-wise into object, relation, and geometry evidence, which naturally maps to "evidence → prescription" style adjustment suggestions.

For scalable learning, we cast structure induction as penalized maximum likelihood and employ an EM-type block-pursuit algorithm: in the E-step we perform terminal matching, relation instantiation, and geometric evaluation; in the M-step we add OR branches or refine constraints by penalized

---

*Corresponding author: jinxin@besti.edu.cn / jinxin@bigai.ai

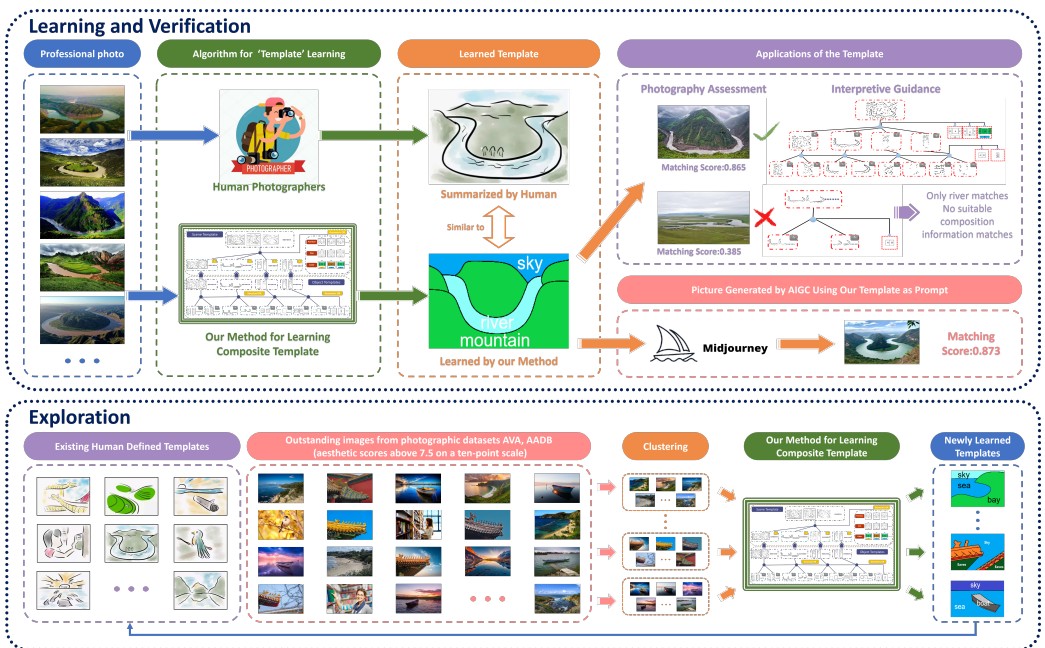

Figure 1: Overview of the compositional AND–OR template framework (top: Learning & Verification; bottom: Exploration). Top: Learn AND–OR templates from professional photos; use them for consistency scoring, parse-tree guidance, and as structural conditions for AIGC. Bottom: Clustering and semi-supervised expansion yield new theme templates.

marginal gain, while local mutual exclusivity and sparsity regularization suppress combinatorial explosion. Learning proceeds in two levels: we first obtain reusable object templates, then reuse them as scene terminals to compose scene templates. We further introduce a semi-supervised extension: a dual threshold of matching gain + structural consistency bootstraps structural increments from unlabeled data, with conflict resolution and early stopping for robust expansion.

We use a licensed collection of professional photographers' works as small-sample seeds and an unlabeled expansion pool: the former initializes object/scene template learning, the latter supports semi-supervised growth. On this basis, we validate the method as a representation-learning framework, including downstream applications: photography guidance, AIGC generation guidance and film-poster layout design, used respectively for framing suggestions, re-ranking generation results, and layout optimization with the template score as the objective. We also evaluate on existing aesthetic corpora like AVA and AADB (Li et al., 2010).

The main contributions of our paper can be summarized as follows:

- **Compositional representation and unified scoring:** We extend AND–OR templates from the *object* level to the *scene* level. Within a maximum-entropy log-linear model we define a *unified consistency score* (log-likelihood gain) that supports training selection and test evaluation, and that *decomposes* into object / relation / geometry terms.

- **Penalized MLE with EM-type block-pursuit:** We instantiate learning as penalized maximum likelihood with an EM-type block-pursuit procedure *adapted* to compositional induction. Local mutual-exclusivity and sparsity regularization curb combinatorial growth, and two-level reuse (objects → scenes) improves parameter efficiency.

- **SSE: semi-supervised structure expansion:** A dual-threshold rule (matching-gain + structural-consistency) bootstraps new branches from unlabeled images, with conflict resolution and early stopping for robustness.

- **Faithful interpretability and prescriptive guidance:** The score's term-wise factorization provides *evidence-to-decision* attribution; parse graphs visualize activated terminals and (dis)satisfied constraints and translate attributions into actionable structure/geometry edits.

## 2 RELATED WORK

### 2.1 COMPOSITIONAL AND GRAMMAR-BASED VISION

A long tradition explains images by *parts* and their *relations*. Classical formulations include deformable templates (Yuille et al., 1992), Active Appearance Models (Baker et al., 2004), pictorial structures (Felzenszwalb & Huttenlocher, 2005), constellation models (Fergus et al., 2007), hierarchical parts dictionaries (Fidler & Leonardis, 2007), and Active Basis (Wu et al., 2010). Discriminatively trained part-based detectors (DPM) established that coupling appearance with geometric deformation is highly effective for robust detection (Felzenszwalb et al., 2009). Beyond single categories, taxonomies and recursive compositionality share parts/appearance across classes and views for better sample efficiency (Ahuja & Todorovic, 2007; Detection et al., 2010).

Within grammar-based modeling, *Hybrid Image Templates* (HIT) integrate sketch primitives with texture/flatness/color cues and are learned by information projection (Si & Zhu, 2011). The *AND–OR Template* (AOT) makes reconfigurability explicit by separating **AND**-nodes (joint composition/constraints) from **OR**-nodes (structural/geometric alternatives), yielding a template that enumerates valid configurations while remaining interpretable (Si & Zhu, 2013). From a broader theoretical standpoint, stochastic/attributed image grammars provide a top–down/bottom–up parsing view in which terminals contribute local evidence and higher nodes regularize legal structures (Zhu et al., 2007; Liu et al., 2014; Park et al., 2017). In parallel, representation learning evolved from HOG (Dalal & Triggs, 2005) to deep backbones (VGG, ResNet) and attention mechanisms (SE, ViT) (Simonyan & Zisserman, 2014; He et al., 2016; Hu et al., 2018; Dosovitskiy et al., 2020); while these models excel in accuracy, their internal reasoning is typically opaque at the composition level.

**From objects to scenes.** Compared with prior AOT/HIT usage that primarily targets object categories, we *extend AOT from the object level to the scene level*. Concretely, learned object templates are *reused as scene terminals* and composed via AND/OR operations under explicit inter-object constraints, so that scene regularities (e.g., relative placement, scale compatibility) are modeled without proliferating low-level parts. This design aligns with attributed grammar–based scene parsing (Liu et al., 2014) and is compatible with object–relation–attribute structure widely adopted in scene understanding corpora (Krishna et al., 2017; Xu et al., 2017). The result is a two-level, interpretable representation in which object knowledge transfers upward to structure scenes.

### 2.2 STRUCTURE-AWARE SCORING AND UNIFIED LEARNING

Structure-aware quality modeling progressed from subject/semantic cues (Luo & Tang, 2008) to large-scale learning setups (e.g., AVA) that support quantitative evaluation and correlation with human preference (Murray et al., 2012), and to explicitly reasoning about recurring compositional patterns (Nguyen et al., 2018). Our framework is theory-leaning: under a *maximum-entropy log-linear* model, we define a unified *consistency score* as log-likelihood gain against a reference distribution. The score *decomposes* into object, relation, and geometry terms—mirroring the grammar—so that positive/negative evidence aggregates transparently to a final rating. This view connects to classical maximum-entropy feature binding and Markov random field modeling (Zhu et al., 1998), while remaining faithful to modern compositional parsing.

Methodologically, we pursue *scalable structure induction* by penalized estimation with sparsity and local mutual exclusivity, alternating between bottom–up terminal matching and top–down constraint refinement. The two-level reuse (objects $\rightarrow$ scenes) curbs combinatorics, and the resulting consistency score provides term-wise attribution that can be used for principled selection, comparison, and ablation. In contrast to black-box predictors (Simonyan & Zisserman, 2014; He et al., 2016; Dosovitskiy et al., 2020), the proposed family preserves explicit *part–relation–geometry* factors and yields an auditable path from evidence to decision.

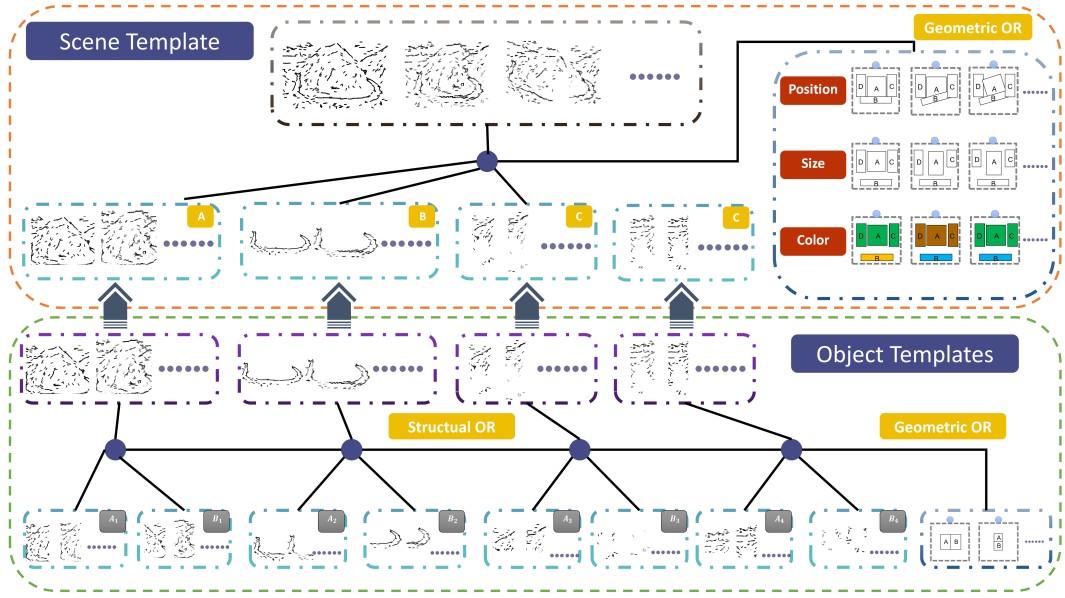

Figure 2: Detailed illustration of the compositional AND–OR template. OR nodes enumerate structural variants and geometric transforms; AND nodes (solid blue circles) encode joint satisfaction of their children; terminal nodes correspond to observable features.

## 3 COMPOSITIONAL TEMPLATE REPRESENTATION

### 3.1 TASK AND DATA

We use a small authorized set of professional photographs as seeds and maintain an unlabeled pool for semi-supervised expansion. The goal is to induce *compositional* (reusable) structured templates from thematically coherent image collections, and to derive a unified scoring rule as well as evidence-to-prescription suggestions. Data provenance and statistics are deferred to **Appx. A.1**.

### 3.2 COMPOSITIONAL AND–OR TEMPLATES AND UNIFIED SCORING

#### 3.2.1 TEMPLATE STRUCTURE

We adopt a two-level representation composed of **object templates** and **scene templates**. At the object level we use an AND–OR graph: **AND** nodes encode relational and geometric constraints that must be satisfied jointly; **OR** nodes capture structural and geometric variants; **terminals** are observable primitives with statistical attributes. At the scene level, learned object templates are reused as "scene terminals" and combined via AND–OR compositions to encode inter-object constraints and equivalent variants. As an illustrative example, consider a "river surrounding mountains" scene with four main objects: river (B), middle mountain (A), left mountain (C), and right mountain (D). The root is an OR node that enumerates valid scene configurations; its children are AND combinations of structural and geometric settings for each component. Further down the hierarchy, the middle mountain and the river can be split into left/right segments, and the left/right mountains into upper/lower segments. These segments serve as terminals in object templates, whereas scene templates treat object templates as terminals to avoid configuration blow-up if tiny segments were elevated to scene terminals. Fig. 2 visualizes the structure: terminals (light blue rectangles), AND nodes (solid blue circles), and OR variants (dashed boxes).

Both levels are regulated by a stochastic context-free grammar (SCFG) to compactly model structural & geometric variability and higher-order part interactions. Terminals are represented by *Photography Art Templates* (PATs), consisting of image primitives and histogram descriptors (e.g., color/texture), each carrying position, scale, and orientation attributes.

A PAT can be specified by a list:

$$\text{PAT} = \{(B_1, x_1, y_1, s_1, o_1), (h_2, x_2, y_2), (h_3, x_3, y_3), (B_4, x_4, y_4, s_4, o_4), ...\}$$

where $B_1$, $B_4$, ... are image primitives and $h_2$, $h_3$, ... are histogram descriptors for texture, flatness and color. $(x_j, y_j)$ denote the selected locations and $o_j$ are the selected orientations.

### 3.2.2 UNIFIED SCORING

Let $\chi_+ = \{I_1, \ldots, I_N\}$ be positive images drawn from a target distribution $f(I)$ and let $q(I)$ denote a reference distribution over generic natural images. Starting from $q$, we progressively reweight by selected features toward $f$,

$$q(I) = p_0(I) \to p_1(I) \to \cdots \to p_T(I) = p(I) \approx f(I). \tag{1}$$

Under the maximum-entropy principle, the model admits a log-linear form

$$p(I) = q(I) \prod_{t=1}^{T} \left[ \frac{1}{z_t} \exp\{\beta_t r_t(I)\} \right], \tag{2}$$

where $\beta_t$ parameterizes feature $r_t$, and $z_t = \sum_{r_t} q(r_t) \exp\{\beta_t r_t\}$ is the normalizer. Given a template Temp and a configuration $(\mathbf{s}, \mathbf{g})$ (structural and geometric), the complete likelihood is

$$p(I, \mathbf{s}, \mathbf{g} \mid \text{Temp}, \beta) = p(\mathbf{s}, \mathbf{g} \mid \text{Temp}) \cdot p(I \mid \mathbf{s}, \mathbf{g}, \beta), \tag{3}$$

with the conditional likelihood

$$p(I \mid \mathbf{s}, \mathbf{g}, \beta) = \exp\left\{ \sum_{k=1}^{K} s_k \left( \sum_{j=1}^{D} \beta_{k,j} r_j(I) - \log Z_k \right) \right\} q(I), \tag{4}$$

Here $D$ is the number of candidate features and $K$ is the number of template terminals, $\beta$ is the parameter for the t-th selected feature $r_t$ and $z_t$ ($z_t \geq 0$) is the individual normalization constant determined by $\beta$. q(I) represents the reference distribution.

We define the **template matching score**, also known as the consistency score and the information gain, as the log-likelihood ratio.

$$\text{Score}(I) = \log \frac{p(I \mid \mathbf{s}, \mathbf{g}, \beta)}{q(I)} = \sum_{k=1}^{K} \text{Score}(\text{PAT}_k, I), \tag{5}$$

where each activated terminal contributes

$$\text{Score}(\text{PAT}_k, I) = s_k \left( \sum_{j=1}^{D} \beta_{k,j} r_j(I) - \log Z_k \right). \tag{6}$$

This additive decomposition provides a direct mapping from *evidence* to *prescription*: positive/negative evidences at the part or relation level translate into actionable geometric or structural adjustments.

## 4 LEARNING COMPOSITIONAL AND–OR TEMPLATES

This section details the EM-type block-pursuit procedure for learning AND–OR templates under the unified scoring and log-linear modeling introduced earlier. The learner alternates between estimating latent structural and geometric configurations (E-step) and performing penalized maximum-likelihood updates of structure and parameters (M-step). We first learn object templates and then reuse them as *scene terminals* to induce scene templates. Finally, we describe a semi-supervised extension that enlarges structure from an unlabeled pool via a dual-threshold criterion (matching gain and structural consistency) with conflict resolution and early stopping.

## 4.1 EM-TYPE BLOCK-PURSUIT

### 4.1.1 TRACKING MATRIX (E-STEP)

We begin by applying YOLOv11 to detect and segment objects, obtaining object-level training samples. For each sample, candidate feature responses are computed to form the data matrix $R \in \mathbb{R}^{N \times D}$, where each entry is $R_{ij} = r_j(I_i) \in [0, 1]$. The E-step, given the current template and parameters, performs terminal matching, relation instantiation, and geometric evaluation. Concretely, within local windows we conduct a finite search over translation, scale, and rotation; per-terminal contributions are evaluated using the additive matching score, and only combinations that satisfy the AND-node relational/geometric constraints are retained. The E-step outputs sample-wise activated blocks and their poses, which serve as sufficient statistics for the M-step.

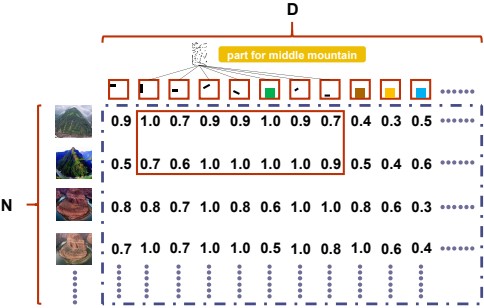

Figure 3: Initial Data Matrix. Data matrix $R$ is a matrix with $N$ (number of positive examples) rows and $D$ (number of all candidate features) columns, and each entry $R_{ij} = r_j(I_i)$ is a feature response ($0 \le R_{ij} \le 1$). Larger value of $R_{ij}$ means feature $j$ appears in image $I_i$ with higher probability.

### 4.1.2 PENALIZED MARGINAL GAIN (M-STEP)

On $R$, we pursue *large blocks* $\{B_k\}_{k=1}^K$ comprised of many high responses; they correspond to frequently occurring, high-confidence $\text{PAT}_k$. A block is specified by a set of shared features (columns) over a subset of examples (rows). Its significance is measured by the summed score inside the block:

$$\text{Score}(B_k) = \sum_{\substack{i \in \text{rows}(B_k) \\ j \in \text{cols}(B_k)}} \left( \beta_{k,j} R_{i,j} - \log z_{k,j} \right). \tag{7}$$

As in the main text, $\text{cols}(B_k)$ are the selected features in $\text{PAT}_k$, $\text{rows}(B_k)$ are the examples on which $\text{PAT}_k$ is activated, and $z_{k,j}$ follows the normalization used for the log-linear form. Maximizing the total block score corresponds to penalized maximum-likelihood estimation; we minimize a two-term objective

$$-L(R, \beta, \mathbf{s}, \mathbf{g}) + \text{penalty}(\beta)$$

with sparsity and *local mutual exclusivity* to suppress overlap and redundancy. In practice, (i) $\beta_{k,:}$ is constrained within local object windows to avoid cross-region leakage; (ii) per image and per local region, at most one block is allowed to be active; and (iii) low-scoring blocks are pruned after ranking. Initialization uses high-frequency terminals and stable local relations. We stop when the marginal information gain of the newly proposed block falls below a threshold, or when both structure size and validation score improvements are below $\epsilon$ for several rounds. The end-to-end pipeline is shown in Fig. 4.

**Algorithmic overview.** For completeness, we provide the full pseudocode of our learning and inference procedures in the **Appx. A**. Algorithm 1 details the *EM-type block–pursuit* procedure used to learn the compositional AND–OR templates from the response matrix $R$ under the log-linear model in Sec. 3, 4. Algorithm 2 specifies the *recursive SUM–MAX inference* used at test time to obtain the structural configuration $s$, geometric configuration $g$, and the decomposable consistency score. The main text reports high-level intuition and ablations; the Appendix gives all steps needed to reproduce results.

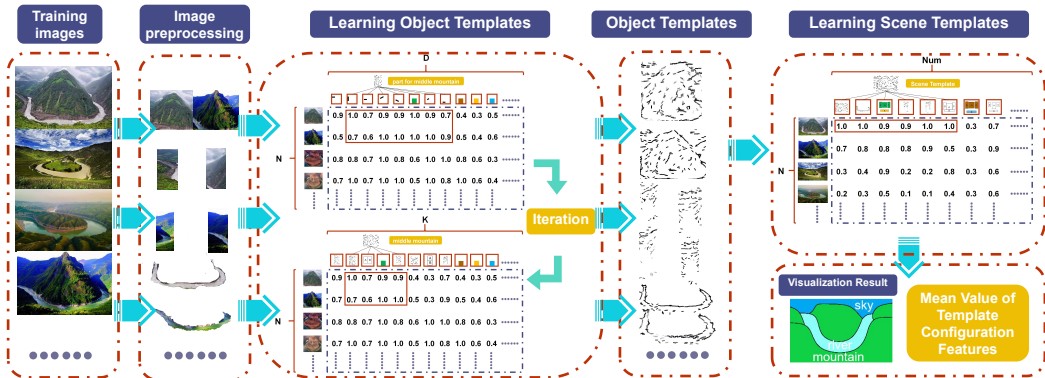

Figure 4: End-to-end template-learning pipeline. Images are first preprocessed by an object detector and segmenter (e.g., YOLOv11). An EM-type block-pursuit algorithm then learns terminal and non-terminal nodes from the data matrix $R$. Learned object templates are composed into scene templates via the same procedure, and mean configuration features are aggregated for visualization.

## 4.2 Two-Level Learning Pipeline

**Object template learning.** On the object-level matrix $R$, EM-type block pursuit iteratively selects high-scoring blocks to instantiate terminals and non-terminals and to estimate the coefficient matrix $\beta^{(T)}$. Sparsity and local mutual exclusivity control model complexity; blocks are ranked by information gain and truncated, yielding a compact and interpretable AND–OR structure at the object level.

**Scene template learning.** In the object-level feature space, we construct $R' \in \mathbb{R}^{N' \times D'}$ where each row corresponds to a photograph in $\chi_+$ and each column to a candidate object-level feature, with entries $R'_{ij} = r'_j(I'_i) \in [0, 1]$. Treating learned object templates as *scene terminals*, we apply the same EM-type block pursuit on $R'$: the E-step emphasizes inter-object relations and global geometric consistency, while the M-step, under penalties, decides whether to add new OR branches or refine cross-object AND constraints. This two-level scheme reuses object knowledge at the scene level and avoids unnecessary pixel-level combinatorial search.

## 4.3 Semi-supervised Learning of Compositional AND–OR Templates

This paper extends the compositional template learning method into a semi-supervised framework, exploring the capability of algorithms to autonomously derive templates from unlabeled image data. This approach enhances learning efficiency while providing novel insights into the semantic depth and aesthetic principles of images. Building upon the preliminary aesthetic template construction, we analyze the relationship between the number of nodes in the template graph and the training image volume, aiming to evaluate model performance and sensitivity to data variability comprehensively.

Additionally, we assess the robustness of semi-supervised learning in the face of disturbances and anomalous data, examining its impact on model stability. Such analysis reveals the algorithm's adaptability in complex environments, offering more resilient and efficient solutions for image processing. Through systematic experiments and analysis, this study aims to provide a richer theoretical foundation for compositional template learning, contributing both to advancements in the practical application of these algorithms in real-world contexts.For detailed information, please refer to **Appx. A.2**.

## 5 Evidence and Validation

We evaluate the proposed compositional AND–OR template along three dimensions—reliability, interpretability, and performance/data efficiency across tasks. The results show that the template

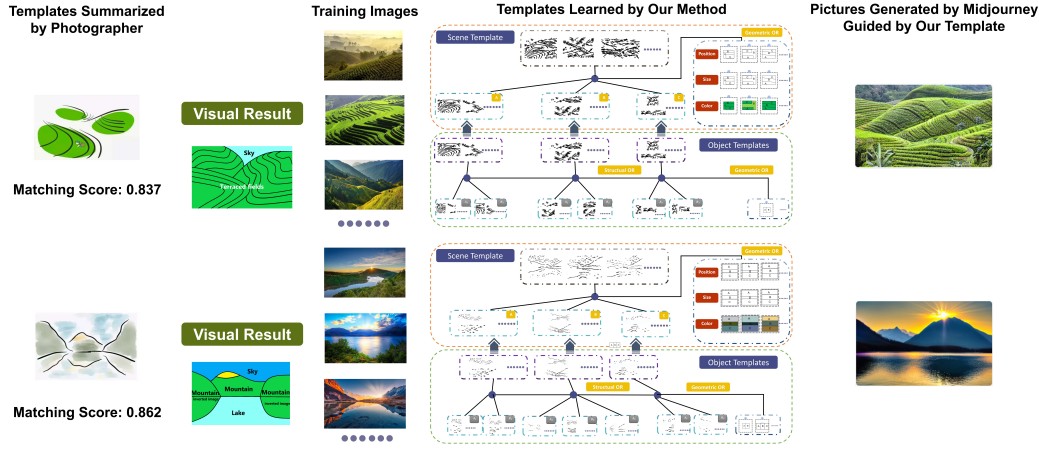

Figure 5: Reliability verification of compositional templates. Expert-summarized templates are compared with templates learned by our method, and effectiveness is further checked using training and generated exemplars. High matching scores correspond to structures preferred by professionals, supporting the reliability and applicability of the learned templates.

consistently captures human-recognized aesthetic regularities, enables explainable predictions of image quality, and achieves performance comparable to or better than deep learning baselines on classification tasks.

## 5.1 CONSISTENCY AND RELIABILITY OF THE COMPOSITIONAL AND–OR TEMPLATE

### 5.1.1 OBJECTIVE CONSISTENCY

Under expert guidance, we curated six representative compositional structure paradigms and, for each paradigm, selected 30 high-quality images to learn a theme template. The learned templates show strong agreement with the expert paradigms in both structural and geometric constraints, indicating that the model automatically distills the same compositional regularities. We quantify alignment using the consistency score defined in equation 5—the log-likelihood gain relative to a reference distribution—and observe scores in a high range, supporting structural consistency. As controls, we report two perturbation studies: (i) counterfactual templates obtained by part substitution or relation shuffling, and (ii) geometric jitter by scale and position perturbations. Both controls lead to marked drops in the consistency score, demonstrating sensitivity to structural and geometric disruptions. To assess structural controllability, We used the learned templates as prompts for an AIGC image generator (e.g., Midjourney); As shown in Fig.5, the outputs adhere to the prescribed part relations and geometric layout, showing that the encoded constraints are executable, rather than mere style reproduction.

### 5.1.2 SUBJECTIVE RELIABILITY

We further conduct a user study with 100 domain practitioners on 10 randomly selected theme templates. Using a five-point scale, participants rate the structural validity and theme alignment (fully consistent, mostly consistent, average, inconsistent, completely inconsistent). As shown in Fig.11, the majority report agreement with professional knowledge: 74.8% fully or mostly consistent, and about 12% inconsistent. Several participants note that the templates capture the layout and inter-element relations emphasized in practice. These subjective results align with the objective analyses and indicate that the learned templates provide a structure prior consistent with human understanding.

Table 1: Quantitative comparison on AVA with single-image input methods.

| Method | Accuracy (%) | Number of Parameters | Training Complexity | SRCC |
|---|---|---|---|---|
| Ours | 85.65 | $\mathbf{2.3 \times 10^3}$ | $\mathbf{6.7 \times 10^4}$ | **0.8419** |
| VGG19 (2-class) | 78.08 | $1.39 \times 10^8$ | $1.96 \times 10^{10}$ | 0.7506 |
| ResNet50 (2-class) | 85.71 | $2.35 \times 10^7$ | $4.09 \times 10^9$ | 0.7842 |
| ViT-B/16 (2-class) | 85.38 | $8.58 \times 10^7$ | $1.75 \times 10^{10}$ | 0.7590 |
| VGG19 + templates | 81.43 | $1.39 \times 10^8$ | $1.96 \times 10^{10}$ | 0.7893 |
| ResNet50 + templates | 85.87 | $2.35 \times 10^7$ | $4.09 \times 10^9$ | 0.8156 |
| ViT-B/16 + templates | **86.20** | $8.58 \times 10^7$ | $1.75 \times 10^{10}$ | 0.7855 |

## 5.2 INTERPRETABILITY AND CONSISTENCY SCORE WITH COMPOSITIONAL AND–OR TEMPLATES

For interpretable scoring and parsing, given a theme template and a target image we compute the consistency score to measure their structural alignment and produce a parse tree for the image. The parse explicitly marks activated terminals and the satisfied or violated relational/geometric constraints. According to equation 5, the score decomposes into object-, relation-, and geometry-level contributions, yielding a clear attribution pathway: satisfying key parts and constraints increases the corresponding contributions and the overall score; missing required parts or violating constraints suppresses activations and reduces the score. Fig. 12 shows typical cases: the template structure (left) and image parses with scores (right). This mechanism supports auditable explanations and provides structured repair suggestions for downstream use. Overall, the objective and subjective evidence, together with interpretable outputs, substantiates our central claim: compositional AND–OR templates reliably capture human-recognized structural regularities, yield a perturbation-sensitive, decomposable consistency score, and act as executable structural constraints for downstream systems.

To further validate the effectiveness of the template scoring, we compared the model prediction with manual subjective evaluation. For detailed information, please refer to the **Appx. A**.

### 5.2.1 PERFORMANCE EVALUATION AND BASELINE COMPARISON

We evaluate the compositional AND–OR template as a structured, interpretable classifier on AVA. The AVA labels serve as proxy supervision to assess the discriminative power and inductive bias of the representation. For each of 14 themes, we select 140 high-score and 140 low-score images to learn two theme templates. At test time, we compute an image's consistency score with each template and predict the class with the higher score. The score is a real-valued alignment metric defined by equation 5 as the log-likelihood gain relative to a reference distribution. It supports ranking, correlation analysis, and thresholding.

Baselines include VGG19 Simonyan & Zisserman (2014), ResNet He et al. (2016) and ViT-B/16 Dosovitskiy et al. (2020). We also perform late fusion by combining template configuration features with deep features to test complementarity. Metrics are Accuracy , FLOPs (floating-point operations, as a proxy for computational cost), and Spearman rank correlation (SRCC) between **the consistency score** and **the AVA annotation scores**.

Table 1 shows that, with substantially less data and far fewer parameters, the compositional template achieves accuracy comparable to strong baselines. Late fusion further improves the deep networks. The consistency score attains higher SRCC with human annotations, indicating stronger alignment between the structured representation and the label standard. Importantly, under the log-linear formulation the score decomposes into object-, relation-, and geometry-level contributions, yielding a clear attribution pathway that links structural evidence to the final decision. This demonstrates a favorable trade-off among discriminative power, data/parameter efficiency, and interpretability.

**Other experiments** In addition to the above verification, we also conducted other objective and subjective experiments, please refer to the **Appx. A** for details

## 6   LIMITATIONS AND CONCLUSION

We introduce a compositional AND–OR template with a maximum-entropy log-linear unified score, learned via penalized MLE with an EM-type block-pursuit and a semi-supervised structural expansion across object and scene levels, achieving competitive performance with strong baselines while preserving interpretability and actionability. Limitations include subjective aesthetics and limited scale blurring theme boundaries, dependence on initial templates and expert curation, lack of systematic cross-cultural/domain tests, coverage of parts/geometry but not lighting/material/portrait factors, hyperparameter sensitivity in the dual-threshold semi-supervised stage, and prototype-level application modules. The impact is an evidence-to-prescription structured paradigm with few-shot data efficiency and transfer, tempered by risks of stylistic/cultural bias and "templatized" creation. Future work will expand and stratify data, incorporate higher-level factors, strengthen evaluation and workflow integration, and conduct larger-scale cross-domain validation.

## ACKNOWLEDGMENT

This paper was supported in part by the National Natural Science Foundation of China under Grant 62476013 & 62072014, and the Opening Project of the State Key Laboratory of General Artificial Intelligence, BIGAI/Peking University, Beijing, China (Project No. SKLAGI2025OP01).

## ETHICS STATEMENT

This work studies compositional representations for art and design. No personally identifiable information is used. The curated datasets (e.g., AVA/AADB and licensed professional photographs) are used under their respective terms; for scanned or third-party images we obtained permission or used publicly available items with proper rights, and we release only non-identifying derivatives. Our user evaluation (expert preference ratings) was conducted with informed consent and without storing personal data; when required, institutional approval was obtained.

## REPRODUCIBILITY STATEMENT

We provide implementation details for the model and learning procedure (Secs. 3, 4), including the scoring definition (Eqs. 2–5), search ranges, and penalties. Dataset composition and splits are described in Appx. A.1; semi-supervised settings in Appx. A.2. Additional plots are included in the A.7. An anonymized code package and instructions are supplied as supplementary materials.

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

## A APPENDIX

### A.1 COMPOSITIONAL IMAGE TEMPLATE DATASET (CITD).

To support compositional representation and AND–OR template learning in art and design, we curate the **Compositional Image AND–OR Template Dataset (CITD)**. The dataset is built from public sources (e.g., AVA, AADB) and expanded with images collected from professional photography websites, works by renowned photographers, and scans from photography monographs. Images are grouped by *compositional structure paradigms*—for example, "U-shaped bay", "boardwalk", "flying eaves", and "skyscraper atrium patterns"—so that parts, relations, and geometric constraints can be learned and evaluated at the theme level.

CITD contains **750** images across **15** compositional themes, with **50** images per theme (see Fig. 6 for examples). For each theme, we designate **525** images as *structure exemplars* that clearly instantiate the target composition, and **225** images as *non-exemplar controls* sourced from popular image-sharing platforms and web search results under the same theme tags. The dataset provides theme-level labels only (no pixel-level annotation), and is intended to (i) seed AND–OR template learning with positive examples, (ii) supply controls for calibration and robustness, and (iii) offer a unified basis for consistency scoring and downstream evaluations.

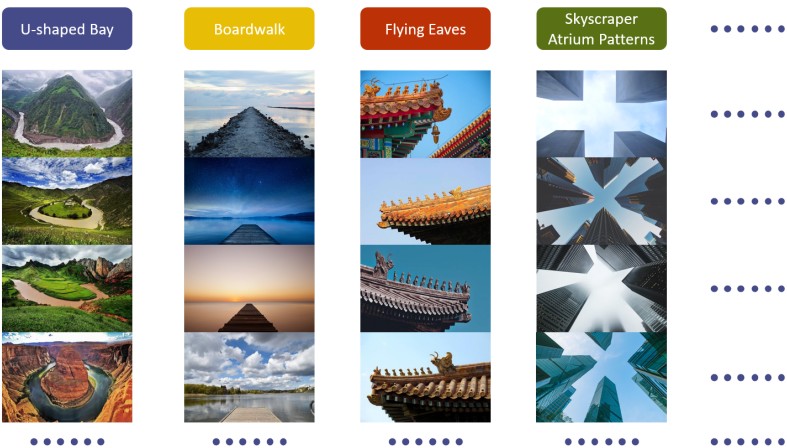

Figure 6: Partial display of our Compositional Image Template Dataset.

### A.2 DETAILS OF SEMI-SUPERVISED LEARNING

### A.2.1 TRAINING DETAILS

As established aesthetic datasets, the AVA and AADB datasets contain a wealth of multi-theme and multi-dimensional aesthetic images, making them highly suitable as foundational data sources for the automatic discovery of templates (Murray et al., 2012; Li et al., 2010). This section proposes a semi-supervised template learning method to learn new image templates on the AVA and AADB datasets.

First, we manually select aesthetic images of the same theme from the datasets to construct the corresponding initial templates, serving as the starting point for semi-supervised learning. Next, the algorithm automatically selects images from the dataset for learning. When faced with training samples of a new theme, the algorithm first attempts to interpret them using the existing templates (Si & Zhu, 2013; Wu et al., 2010). If the matching score between the existing templates and the new images is low, we integrate the new configurations and update the existing templates. By iteratively refining this process, we can gradually enrich the content of the initial templates, ultimately forming a compositional AND–OR template graph that includes multiple templates. The detailed workflow of the semi-supervised learning process is illustrated in Fig. 7.

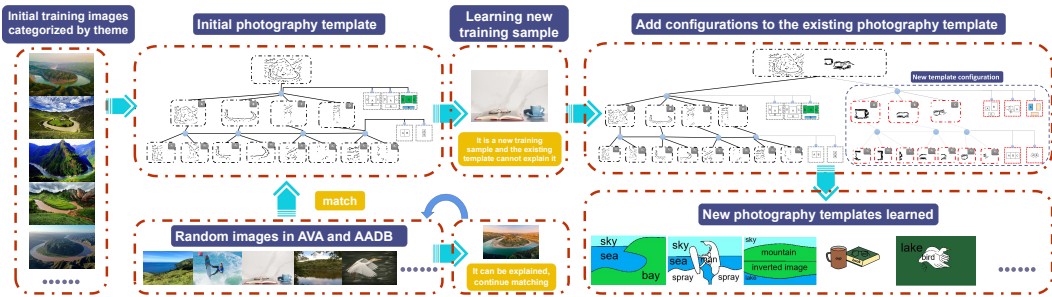

Figure 7: Semi-supervised Compositional AND–OR Template Learning Process. Starting from an initial template, the algorithm autonomously selects images from the AVA and AADB datasets for further learning. When new thematic samples are encountered, the algorithm attempts to match them with existing templates; if the match score is low, new configurations are added to the existing template to update it. Through iterative refinement, the content of the template is enriched, eventually forming a compositional AND–OR graph encompassing multiple styles. This process enhances the diversity of the templates while improving the flexibility and efficiency of aesthetic analysis through semi-supervised learning.

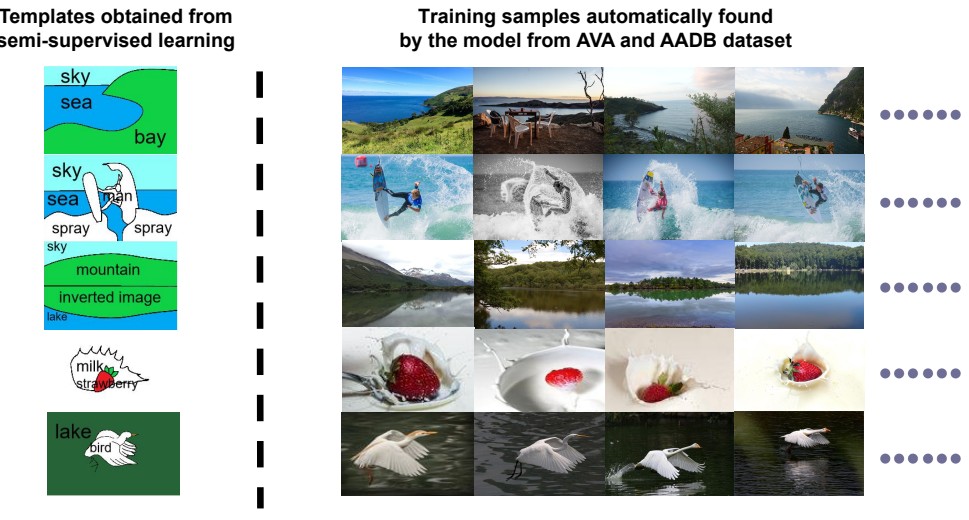

Figure 8: New templates obtained from semi-supervised learning.

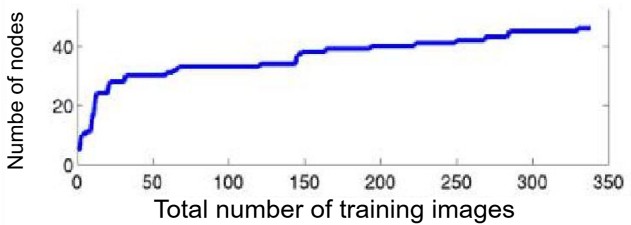

Figure 9: The relationship between the number of training images and the growth of the number of nodes.

### A.2.2 PERFORMANCE ANALYSIS

Compared to supervised learning methods, semi-supervised image aesthetic template learning faces a larger and more complex volume of data, and autonomous learning implies varying quality of the acquired training data Fergus et al. (2007). Therefore, this section primarily focuses on two aspects of performance evaluation: first, the growth of the template AND–OR graph when handling large datasets; and second, the robustness of the learning method when confronted with unrelated training data. The former reflects the capacity of the learning method to handle a substantial influx of data autonomously, while the latter demonstrates its resilience against anomalous or noisy data.

In analyzing the growth of the AND–OR graph under semi-supervised template learning, we examined the relationship between the number of nodes in the template AND–OR graph and the total number of training images, as shown in Figure 9. The results indicate that the number of nodes in the graph exhibits sub-linear growth, suggesting that many nodes are highly replicable and can be shared across multiple subgraphs. By employing the methodologies constructed in **Secs. 4**, we can automatically optimize these redundant nodes to effectively address the challenges posed by large-scale training data Simonyan & Zisserman (2014).

To validate the robustness of the semi-supervised learning method, this section designs a cross-validation experiment. First, two independent training samples of size $n$ are selected from the same thematic training dataset, and corresponding templates are learned. Subsequently, the KL divergence is calculated on a third independent sample of the same theme (test data) to assess the differences between the two templates. The specific calculation method is as follows:

$$K(TMP^*|TMP) \approx \sum_{i=1}^{m} \log \frac{p(b_i; TMP^*)}{p(b_i; TMP)}$$

where TMP* and TMP represent the two trained templates. A smaller calculated KL divergence value demonstrates a lesser degree of difference between the two templates, thus reflecting the reliability of the semi-supervised learning algorithm under random training data Baker et al. (2004). We repeatedly observed the changes in KL divergence for different sample sizes $n$, with the results illustrated in Figure 10. The experiments demonstrate that when $n$ reaches 100, the KL divergence of the templates obtained through semi-supervised learning can be reduced to 0.1, indicating good robustness.

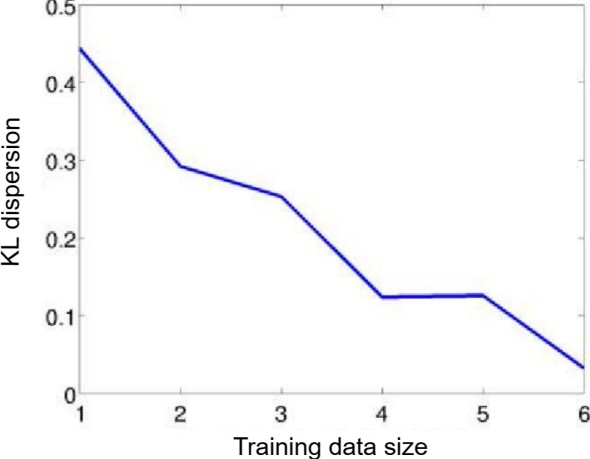

Figure 10: Changes in training data size and KL divergence.

### A.3 HUMAN ALIGNMENT

To assess whether the *compositional AND–OR templates* provide a structural representation aligned with human judgments, we conducted a pairwise-preference study. We randomly sampled 50 images to form a set $P$. For each image $P \in P$, we paired it with every other image in the set, yielding

**2**,**500** binary comparisons completed by **15** raters with art/design backgrounds. Pairwise comparison, rather than absolute scoring, is more in line with expert practice and more stably reflects *structural* differences.

For each image $P$, let the binary variable $y_{P,J} \in \{0,1\}$ indicate whether $P$ is preferred over $J$. We estimate the *win probability*

$$\text{Score}(P) = \mathbb{E}_J\big[\mathbf{1}\{P \text{ wins against } J\}\big], \tag{8}$$

and fit a logistic/Bradley–Terry model to the win–loss outcomes to obtain a *latent human score* $\hat{u}(P) \in (0,1)$:

$$\text{logit}\big(\hat{u}(P)\big) = \lambda_0 + \sum_k \lambda_k\, r_k(P), \tag{9}$$

where $r_k(P)$ can be aligned with the template's consistency terms (e.g., object-, relation-, and geometry-level cues).

We then compare $\hat{u}(P)$ with the *consistency score $S(P)$* defined in equation 5—the log-likelihood gain relative to a reference distribution, normalized to $[0,1]$—and measure proximity via mean squared error (MSE). As shown in Fig. 16 (dashed line indicates ideal alignment; MSE = 0.0286), the consistency score exhibits a stable ordering consistent with human judgments, with small deviations.

These results indicate that, within the structural paradigm of art and design, the unified score produced by compositional AND–OR templates aligns well with human assessments of *part–relation–geometry* organization. Overall, this study validates our *representation–scoring* framework from a human-alignment perspective in art and design settings.

## A.4 STRUCTURE-AWARE ASSESSMENT AND GUIDANCE FOR PHOTOGRAPHY

In the field of photography, professional photographers often find commonalities and extract repeated structural patterns from a series of photographs with the same theme, thus forming some "template" techniques for photography. Beginners in photography can refer to the "templates" summarized by these experts, imitate shooting, and add their own thinking on this basis to create better works. We use a compositional AND–OR template to achieve this operation, aligning the input photo with the theme template and resolving issues such as missing objects, poor scene size and orientation resulting in poor composition in the photo. At the same time, the output includes both textual and visual actionable guidance, guiding photographers on how to improve on the shortcomings of their photographic images.

As shown in Fig. 13, we have designed a rule corpus that will output corresponding guidance rule statements based on the comparison results of two templates. At the same time, it will also output combinable AND–OR images, interpreting the image by indicating which part is activated. Finally, provide corresponding guidance on the original image based on the visualization results of the template.

## A.5 TEMPLATE-CONSTRAINED POSTER GENERATION

We further test structural controllability in an AIGC workflow for film posters. From poster examples, we learn a theme template and use it as a conditioning signal for a general generator. The goal is to enforce part relations and geometric layout specified by the template. As shown in Fig. 14, the generator produces plausible posters that respect the prescribed structure. We also note current limitations: industrial poster design involves complex pipelines, and today's generative models offer limited controllability. Our results are a proof of concept that the learned template acts as a lightweight structural constraint for creative generation.

## A.6 SCENE CLASSIFICATION WITH AND–OR TEMPLATES

To examine transfer beyond aesthetics, we apply the templates to scene classification.

**CITD evaluation.** On a 15-theme dataset, we perform one-to-all tests with 5×5 cross-validation, comparing against HoG Dalal & Triggs (2005) + linear SVM, part-based SVM, and a pretrained

ResNet. As shown in Fig. 15, the template approach outperforms traditional small-sample baselines and approaches the pretrained deep model, while providing parse-level explanations unavailable to black-box classifiers.

**Places365 subset.** We select 10 categories, learn category templates, and rank test images by their template consistency over the 10 templates to make predictions. Table 2 reports Top-1/Top-5 errors. The template method achieves competitive Top-1 and strong Top-5 performance compared to standard deep baselines, at far lower data and parameter costs and with inherent interpretability. These results indicate a favorable trade-off among accuracy, efficiency, and explanation for compositional AND–OR templates on a general vision task.

Table 2: Scene classification on a 10-class subset of Places365. Errors are in %.

| Method | Top-1 Error | Top-5 Error |
|---|---|---|
| Ours | 23.44 | 8.03 |
| Places-365-CNN Zhou et al. (2017) | 23.27 | 8.48 |
| ResNet He et al. (2016) | 23.35 | 8.61 |
| SENet Hu et al. (2018) | 23.67 | 8.26 |

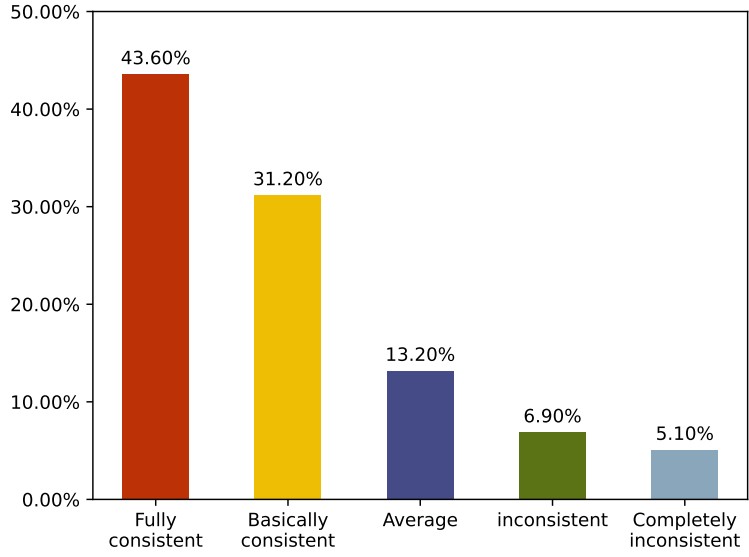

Figure 11: Subjective evaluation of compositional template reliability.

### A.7 OTHER QUALITATIVE EVIDENCE AND PROCEDURAL DETAILS

This section complements the quantitative results with qualitative evidence and end-to-end methodological details. Algorithm 1 specifies the EM-type block–pursuit procedure used for penalized structure induction at both the object and scene levels, and Algorithm 2 details the recursive SUM–MAX inference that yields the structural configuration, geometric pose, and a decomposable consistency score at test time. Figures 17–19 visualize (i) the consistency score and its attribution over part/relation/geometry terms, (ii) reliability checks under counterfactual structural and geometric perturbations, and (iii) representative outputs including parse graphs, activated terminals, (dis)satisfied constraints, and prescriptive edits. Together, these materials make the learning dynamics and the evidence-to-decision pathway explicit, and facilitate reproduction and diagnostic analysis.

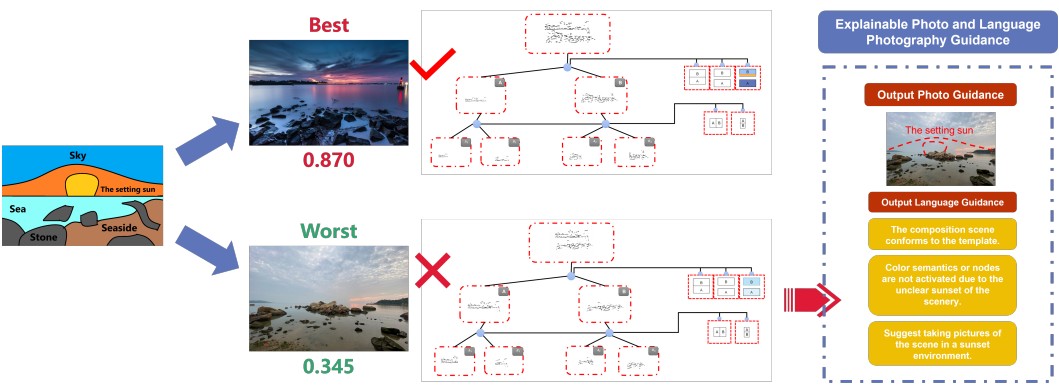

Figure 12: Quantitative assessment results. Besides reporting the consistency score, the method visualizes the parse configuration. For low-scoring images, interpretable guidance is provided by contrasting the parse with the template.

---

**Algorithm 1** EM-type Block–Pursuit Learning of Compositional AND–OR Templates

**Require:** Response matrix $R \in [0,1]^{N \times D}$ (rows: examples, cols: candidate features), initial structural indicator $s^{(0)}$ (optional), maximum #blocks $K$, regularization parameters $\lambda_{\mathrm{mx}}$ (mutual-exclusion) and $\lambda_{\mathrm{sp}}$ (sparsity), tolerance $\varepsilon > 0$ or feature budget.

**Ensure:** Coefficient matrix $\beta \in \mathbb{R}^{K \times D}$, normalizers $Z \in \mathbb{R}_{\geq 0}^{K \times D}$, and learned structural indicator $s$.

1: Initialize $m \leftarrow 0$; set $\beta^{(0)} \leftarrow 0$, $Z^{(0)} \leftarrow 1$, block feature sets $S_k^{(0)} \leftarrow \emptyset$ for $k = 1..K$.
    **repeat**                                                             *// EM-type block pursuit*
2: **E-step (activation & assignment):**
3:    For each example $i$, compute current block scores $q_{i,k} = \sum_{j \in S_k^{(m)}} \beta_{k,j}^{(m)} R_{ij} - \sum_{j \in S_k^{(m)}} \log Z_{k,j}^{(m)}$.
4:    Assign $i$ to its best explaining block $a_i = \arg\max_k q_{i,k}$ (ties broken arbitrarily).
5:    Estimate per-block activation weights $w_{i,k} = \mathbf{1}\{a_i = k\}$ and per-feature empirical means $\bar{r}_{k,j} = \frac{\sum_i w_{i,k} R_{ij}}{\sum_i w_{i,k} + \delta}$.
6: **M-step (feature addition & parameter update):**
7:    For each block $k$ and each candidate feature $j \notin S_k^{(m)}$, compute the *penalized marginal gain*

$$\Delta_{k,j} = \left( \bar{r}_{k,j}\, \hat{\beta}_{k,j} - \log \hat{Z}_{k,j} \right) - \lambda_{\mathrm{mx}} \cdot \mathrm{OVERLAP}(j, S_k^{(m)}) - \lambda_{\mathrm{sp}},$$

   where $\hat{\beta}_{k,j}$ (together with $\hat{Z}_{k,j}$) is the one-dimensional maximum-likelihood update for feature $j$ under the log-linear model.
8:    Select the best feature $j^\star = \arg\max_j \Delta_{k,j}$ for each block $k$.
9: **if** $\max_k \Delta_{k,j^\star} \leq \varepsilon$ **then**
10:     **break**                                                  $\triangleright$ no positive gain remains
11: **end if**
12:    Let $k^\star = \arg\max_k \Delta_{k,j^\star}$; augment $S_{k^\star}^{(m+1)} \leftarrow S_{k^\star}^{(m)} \cup \{j^\star\}$.
13:    Update the corresponding parameters $\beta_{k^\star,j^\star}^{(m+1)} \leftarrow \hat{\beta}_{k^\star,j^\star}$, $Z_{k^\star,j^\star}^{(m+1)} \leftarrow \hat{Z}_{k^\star,j^\star}$; keep other entries unchanged.
14:    Enforce local mutual exclusion and sparsity $(\lambda_{\mathrm{mx}}, \lambda_{\mathrm{sp}})$ by pruning conflicting or weak features.
15: **Re-assignment:**
16:    Recompute $q_{i,k}$ with updated parameters and reassign examples (lines 4–5).
17:    Set $m \leftarrow m + 1$ and continue until tolerance/budget is met.
    **until** convergence
18: Return $\beta = \beta^{(m)}$, $Z = Z^{(m)}$, and the final structural indicator $s$ implied by $\{S_k^{(m)}\}_{k=1}^K$.

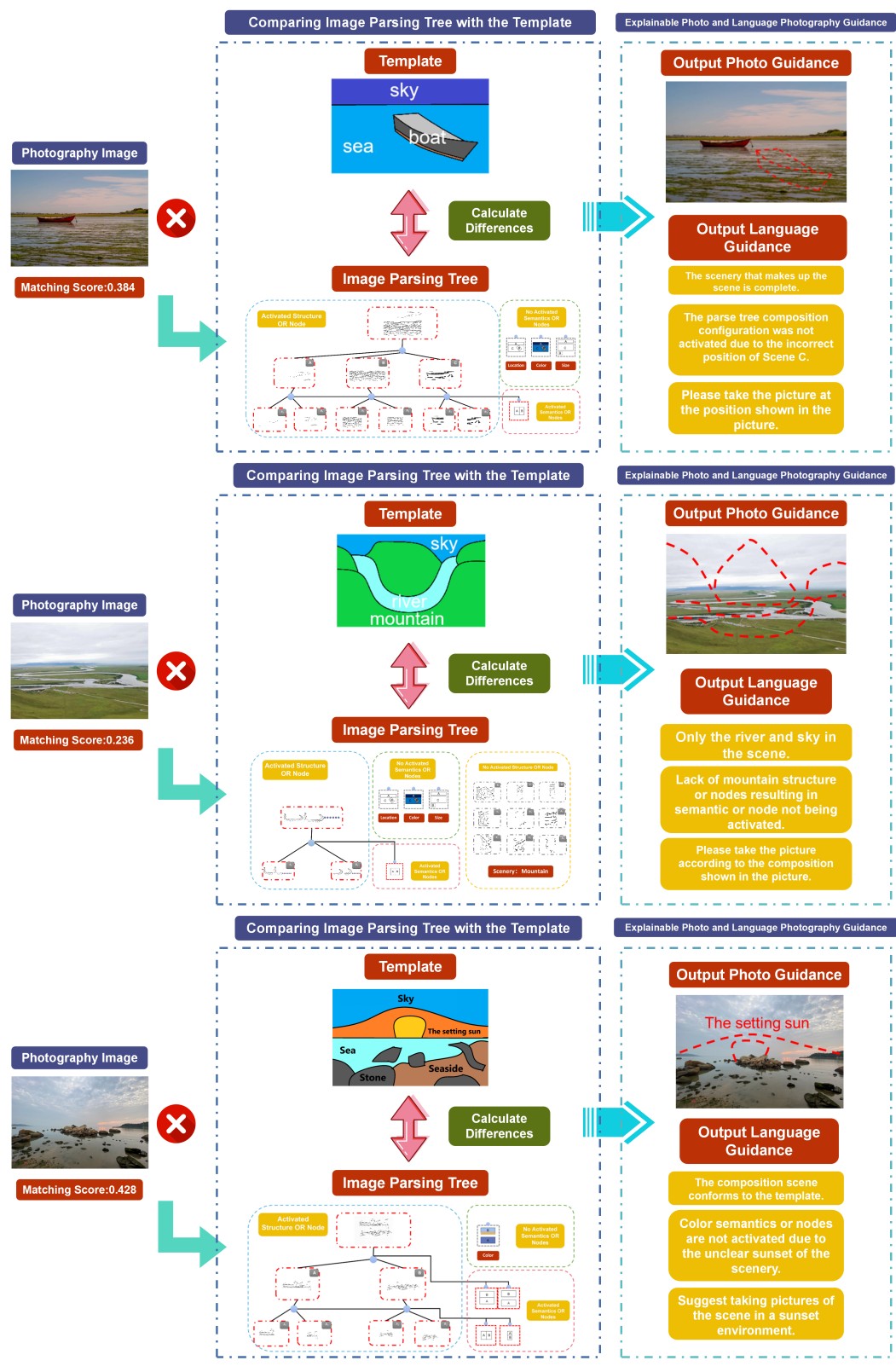

Figure 13: Interpretable evaluation and guidance diagram for photographic images based on compositional AND–OR templates.

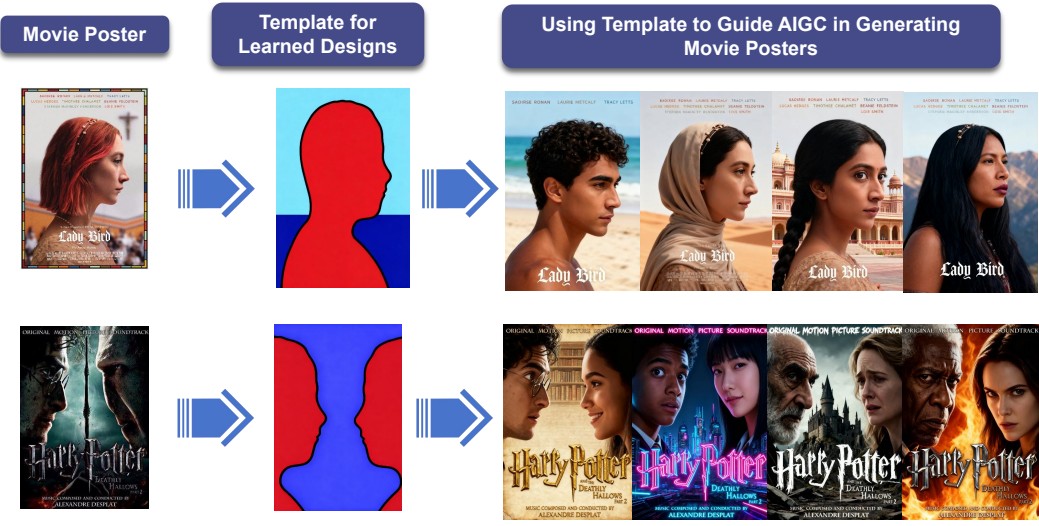

Figure 14: Using the compositional AND–OR template to guide the generation of movie posters.

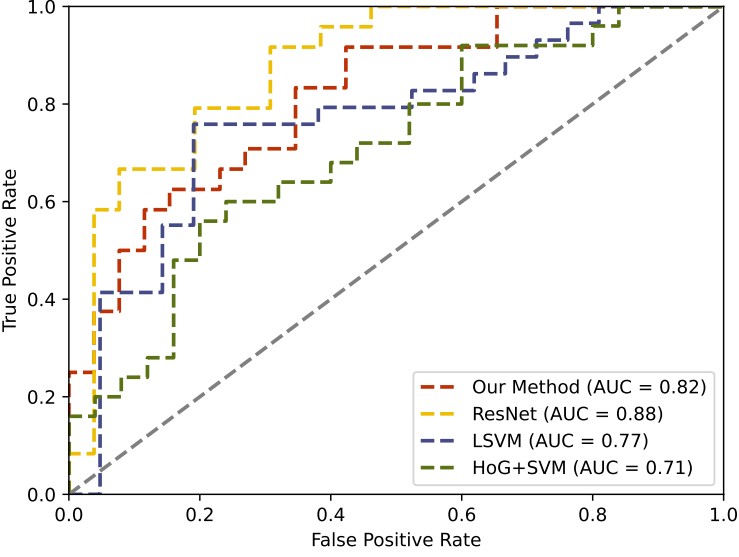

Figure 15: ROC curve of scene classification results.

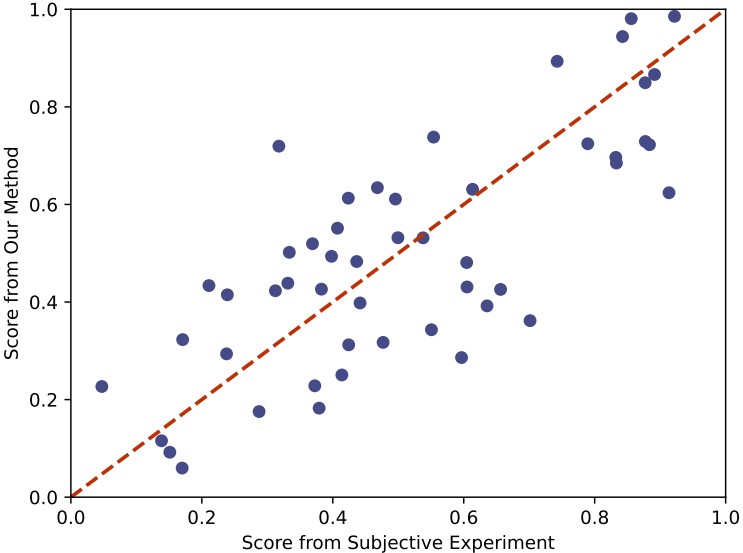

Figure 16: Results of the user study on template reliability.

---

**Algorithm 2** Recursive SUM–MAX Inference (Parsing and Scoring at Test Time)

---

**Require:** Test image $I$; template TMP with topology, terminal sets, coefficients $\beta$ and normalizers $Z$; transformation space $\mathcal{T}$ (positions $x, y$, scales $s$, orientations $o$).

**Ensure:** Structural configuration $s$ (activated parts/objects), geometric configuration $g$ (locations/scales/orientations), and the decomposable consistency score.

1: **Terminal responses (MAX1).** For each terminal feature $j$, compute its response map $r_j(I, t)$ for all $t \in \mathcal{T}$. Optionally allow local jitters $u \in \mathcal{U}$ and record $\text{MAX1}_j(t) = \max_{u \in \mathcal{U}} r_j(I, u \circ t)$ and the argmax $\text{ARGMAX1}_j(t)$.

2: **Part aggregation (SUM2).** For each part $k$ with terminal set $\mathcal{J}(k)$ and each $t \in \mathcal{T}$, compute

$$\text{SUM2}_k(t) = \sum_{j \in \mathcal{J}(k)} \beta_{k,j} \, \text{MAX1}_j(t) \; - \sum_{j \in \mathcal{J}(k)} \log Z_{k,j},$$

and store backpointers to the maximizing terminals.

3: **Object composition (SUM3).** For each object hypothesis $t \in \mathcal{T}$, combine selected parts:

$$\text{SUM3}(t) = \sum_{k \in \mathcal{K}_{\text{active}}} s_k \, \text{SUM2}_k(t),$$

where $s_k \in \{0, 1\}$ encodes structural choices (OR selections, mutual exclusion).

4: **Selection and backtracking.** Find $t^\star = \arg\max_t \text{SUM3}(t)$. Backtrack through the stored pointers (SUM3→SUM2→MAX1) to obtain the activated parts and their transformations, yielding $s$ and $g$.

5: **Output.** Return $s$, $g$, and the score $\text{SUM3}(t^\star)$. The score decomposes additively into object-, relation-, and geometry-level contributions under the log-linear model, enabling per-term attribution.

---

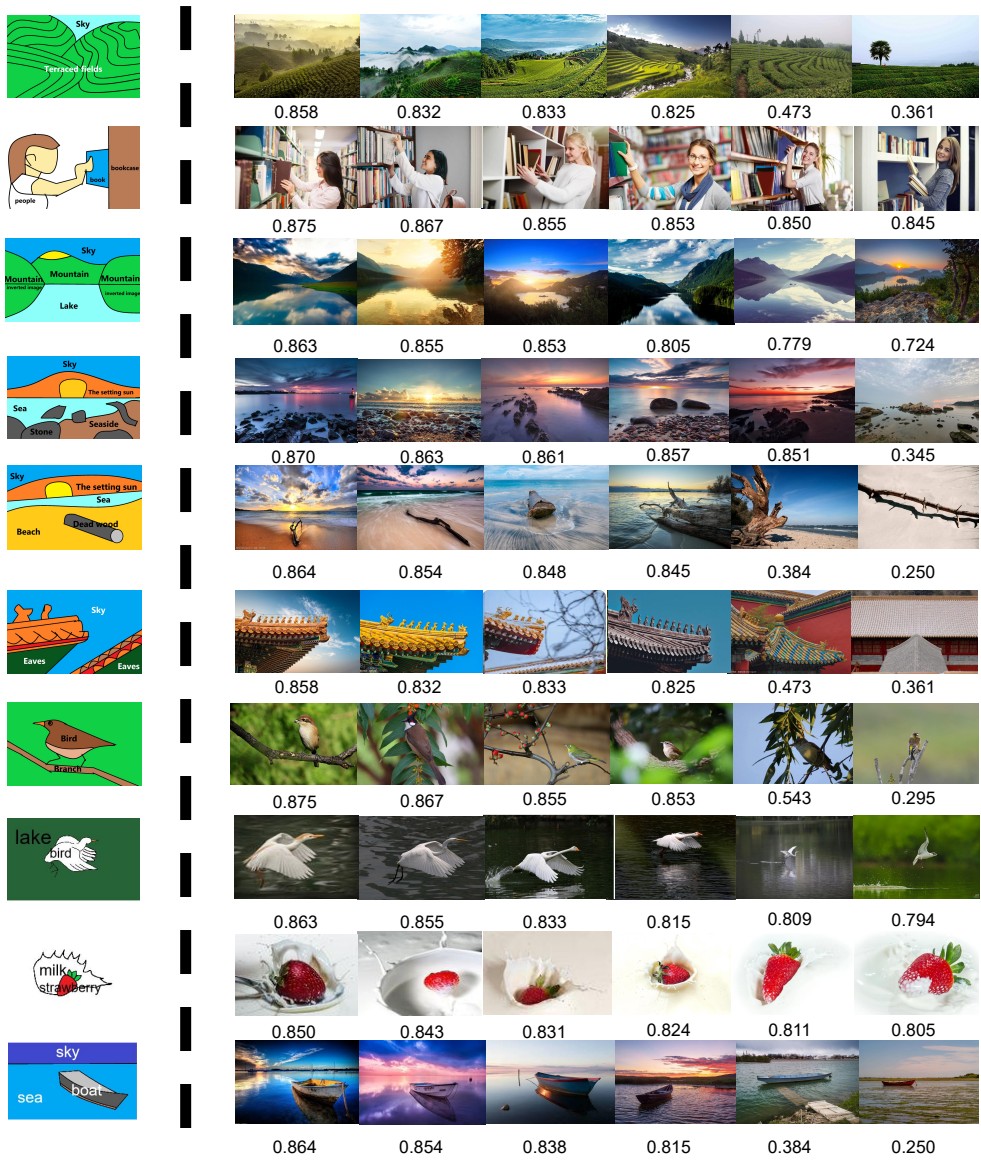

Figure 17: consistency score based on compositional AND–OR templates.

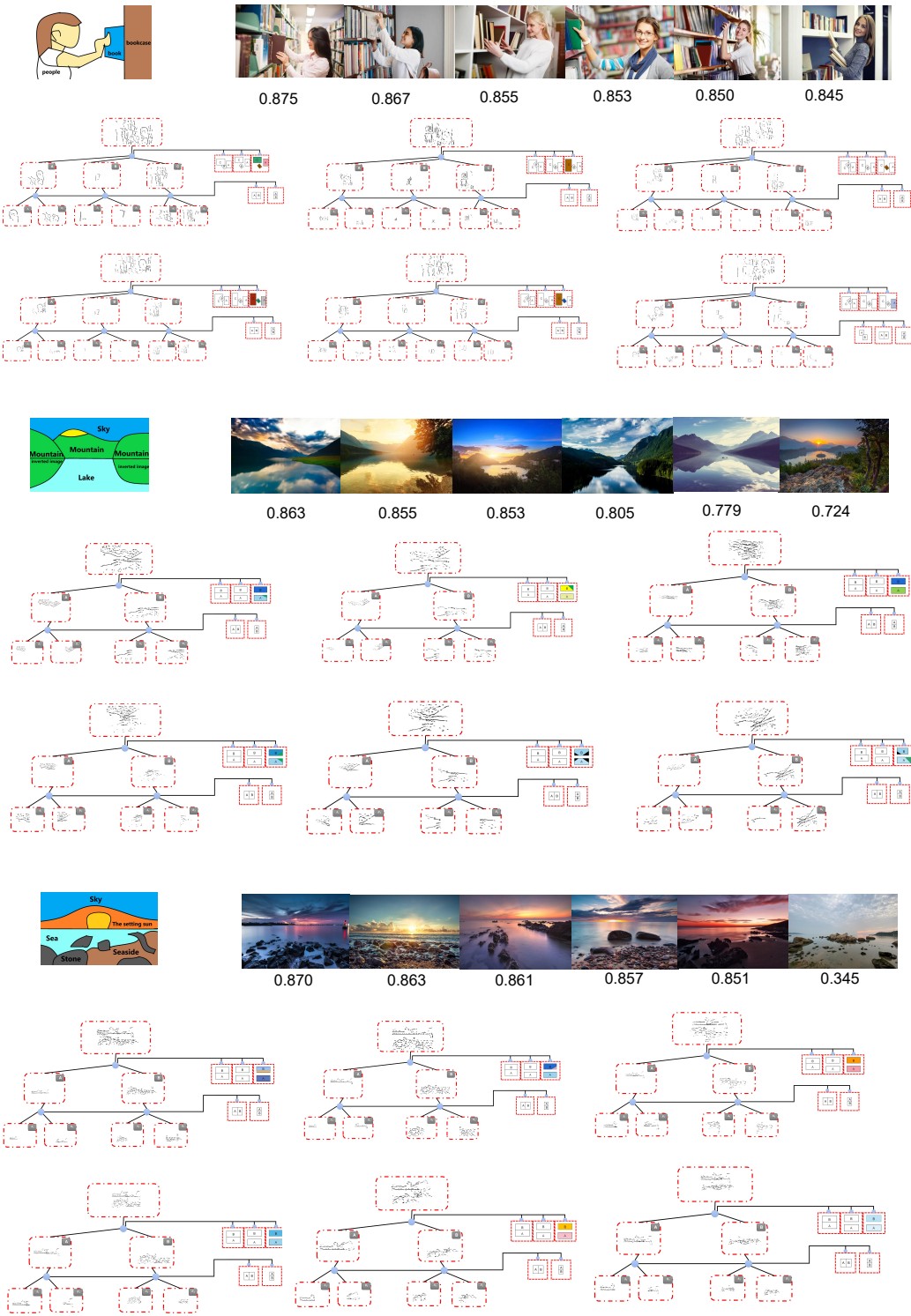

Figure 18: Reliability verification results of compositional AND–OR templates.

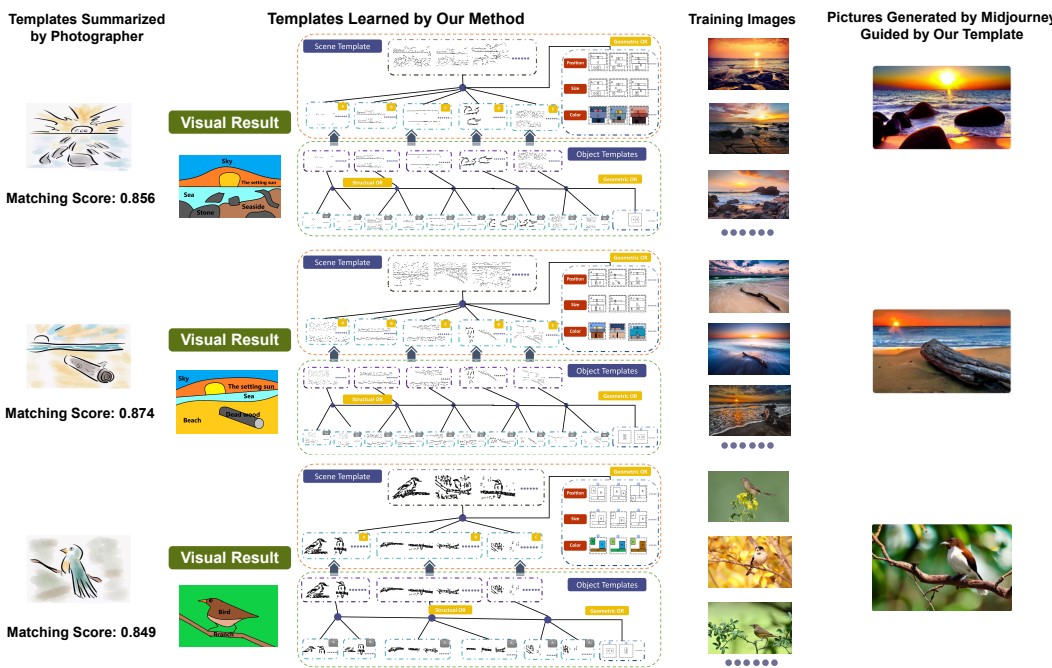

Figure 19: Output results based on compositional AND–OR templates.

## STATEMENT ON THE USE OF LARGE LANGUAGE MODELS (LLMS)

**Scope.** We used an LLM strictly as a general-purpose assistant for *copy-editing and translation* (e.g., abstract, figure captions, and short passages), as well as minor LaTeX/formatting suggestions (citations, line breaks).

**No role in research.** The LLM did *not* contribute to problem formulation, algorithm or model design, implementation, experiments, data or code generation, literature selection, analysis, or conclusion writing.

**Human verification and responsibility.** All LLM-suggested edits were reviewed line-by-line by the authors, with factual checks against the original sources to avoid unverified claims or copyrighted third-party text. The authors take full responsibility for the entire content; the LLM is *not* an author or contributor.

**One-line summary.** The LLM was invited as a *grammar-savvy spellchecker with opinions*, not as a co-author.

