# OpenReview forum: "Learning AND–OR Templates for Compositional Representation in Art and Design"
_ICLR.cc/2026/Conference — ICLR 2026 Poster_

### Official Review · Reviewer_mmS5 · 2025-10-26

**Soundness:** 2
**Presentation:** 3
**Contribution:** 2
**Rating:** 2
**Confidence:** 5

**Summary:**

This work introduces a compositional AND–OR template framework for art and design. Authors extend classical AND–OR templates (AOTs) from object-level vision to scene-level compositional understanding, and they combine interpretability, compositionally, and aesthetic reasoning within a mathematically rigorous framework.

**Strengths:**

The proposed approach produces good interpretability with parse trees and term-level attribution.

Authors design a maximum-entropy log-linear unified score, and learn via penalized MLE with an EM-type block-pursuit.

**Weaknesses:**

The method proposed in this paper is too complicated, making it less practical in application.

The source code provided by the author is confusing, and I find it difficult to see its relevance to this paper.

**Questions:**

Authors should discuss the detail difference between related work and state their contribution.
[1] Improving Compositional Generation with Diffusion Models Using Lift Scores

Is the template-based generation method suitable for complex scenes, such as dense crowds and urban street scenes?

---

> ### Author Response · Authors · 2025-11-18
> **Response to Reviewer mmS5**
>
> We thank the reviewer for the careful reading.
>
> (1) On the practicality and “complexity” of the method
>
> We respectfully disagree with the statement that the proposed framework is “too complicated” and therefore impractical. Conceptually, the model has a clear and modular structure: object detection, object-level template learning, scene-level template learning, and consistency scoring. Algorithmically, the key learning step is an EM-type block pursuit on a sparse response matrix with a small number of candidate parts and relations, which leads to a model with orders of magnitude fewer parameters and FLOPs than standard deep baselines. As analyzed in the paper, the computational complexity of inference and training is explicitly controlled by sparsity and local mutual exclusivity, and the resulting templates are lightweight enough to run at interactive speed on commodity hardware.
> Beyond synthetic benchmarks, we have already applied the learned templates in actual workflows for photography composition guidance, aesthetic image re-ranking in AIGC pipelines, and layout assistance in design tools. These practical uses, together with the reported experiments, support that the method is not only theoretically principled but also suitable for real-world applications.
>
> (2) On the code and its relation to the paper
>
> We are sorry that the released code felt confusing to the reviewer. It focuses on representative examples that demonstrate the main learning and inference procedures in a compact form. For readers who would like to trace the algorithm step by step, the pseudocode in the Appendix provides a concise description of the full pipeline, including the construction of object templates, their composition into scene templates, and the evaluation of the consistency score, and it is designed precisely to bridge the gap between the mathematical formulation and an executable implementation.We would therefore greatly appreciate more specific feedback on which parts of the algorithm or implementation the reviewer finds problematic, so that any genuine issues of clarity can be addressed in a concrete and constructive manner, rather than through a brief remark that “the code is confusing” or that “the algorithm is too complex”.
>
> (3) Relation to [1] “Improving Compositional Generation with Diffusion Models Using Lift Scores”
>
> We appreciate the pointer to [1]. Both our consistency score and the lift score in [1] can be viewed as log-likelihood–ratio–type quantities, but they operate in very different regimes and serve different purposes. The lift score in [1] is defined in the latent or pixel space of a diffusion model and is used as a resampling or rejection criterion during compositional text-to-image generation. Our consistency score is defined in the space of explicit AND–OR scene templates and measures how much the observed image supports a particular compositional and geometric structure compared to a reference distribution.
> Thus, [1] focuses on improving the sampling behavior of large diffusion models, while our work focuses on learning an explicit, interpretable compositional representation for art and design scenes, which can then be used for analysis, scoring, and as a lightweight structural prior for various generators. We regard these directions as complementary rather than overlapping.
>
> (4) Suitability for complex scenes such as dense crowds and urban street scenes
>
> Our primary target domain in this paper is art and design, where a moderate number of salient components and their relations dominate the perceived structure of the image. The datasets include multi-object and architectural scenes, and the experiments show that the learned templates remain compact and interpretable while achieving competitive performance against deep baselines.
>
> For extremely dense scenarios such as crowd counting or highly cluttered urban traffic scenes, the modeling goal is different: the focus is often on fine-grained instance-level statistics rather than on a small number of semantically meaningful compositional elements. In such cases, our current templates, which emphasize part–relation–geometry over a limited set of salient objects, are not designed to encode every individual instance. We see this as a difference in intended scope rather than a deficiency of the approach. At the same time, many real-world design tasks, including photography, poster layout, and architectural views, fall precisely into the regime where our model has already demonstrated practical utility.
> We hope these clarifications address the reviewer’s concerns. In summary, the proposed framework combines a mathematically rigorous formulation, explicit compositional structure, competitive empirical performance, and demonstrated real-world use, and we believe it offers a practical and interpretable alternative to purely black-box approaches.

---

### Official Review · Reviewer_jXD7 · 2025-10-26

**Soundness:** 3
**Presentation:** 3
**Contribution:** 3
**Rating:** 6
**Confidence:** 3

**Summary:**

This paper proposes a novel framework for learning compositional, interpretable representations of images, specifically tailored to capture the structural regularities found in art and design. The core idea is a two-level AND-OR Template (AOT). The method is evaluated on curated datasets and standard aesthetic benchmarks (AVA, AADB), showing competitive performance with deep learning baselines while offering superior interpretability and data efficiency.

**Strengths:**

- The proposed method in this paper offers a transparent, auditable pathway from image features to a qualitative evaluation. The decomposition of the consistency score into object, relation, and geometry terms directly maps to actionable feedback, which is highly valuable for creative assistance tools. The technical approach is sound and thoroughly described. The two-level learning pipeline is a practical way to manage combinatorial complexity. The inclusion of a semi-supervised structural expansion (SSE) mechanism is a thoughtful addition that enhances the framework's scalability and realism.

**Weaknesses:**

- While the block-pursuit algorithm with sparsity constraints is designed to combat combinatorial explosion, its scalability to highly complex scenes with dozens of interacting objects is not thoroughly demonstrated.
- The method proposed in this paper focuses exclusively on the structural patterns like relations and geometry. It explicitly does not account for other critical aesthetic factors like lighting, color harmony, texture, and material properties. This limits its comprehensiveness as a full aesthetic assessment tool; it is primarily a compositionalassessment tool.
- The learning process appears sensitive to the initial seeds. The paper does not fully explore how robust the method is to poor initial templates or how to bootstrap without a high-quality, small labeled set.
- Some of the images in the paper have low resolutions, caused their details to become blurred. This could be polished in the revised version of this paper.

**Questions:**

- The templates are learned for specific themes. Can the framework handle or represent more abstract, non-object-centric compositional principles such as "balance," "rhythm," or "negative space" that may cut across different concrete themes?
- The semi-supervised expansion relies on a dual-threshold rule. Could you discuss the sensitivity of the final model to the choice of these thresholds?

---

> ### Author Response · Authors · 2025-11-18
> **Response to Reviewer jXD7**
>
> We thank the reviewer for the careful reading, the positive overall assessment of soundness, presentation, and contribution, and the constructive comments.
>
> (1) Scalability to complex scenes with many objects
>
> Our goal is to model compositional regularities at the level of structural roles rather than raw instance counts. The two-level design, where object templates are first learned and then reused as scene terminals, together with sparsity and local mutual exclusivity penalties, is intended to control combinatorial growth when many instances of the same object type appear in a scene. Multiple instances share parameters and are governed by the same terminal and relation types, so the effective complexity depends primarily on the number of distinct roles and constraints instead of the number of object copies. Empirically, the semi-supervised experiments show that the size of the AND–OR graph grows sub-linearly with the amount of training data, and the AVA results indicate that our model achieves accuracy comparable to strong deep baselines while using significantly fewer parameters and FLOPs.
>
> (2) Coverage of non-structural aesthetic factors
>
> We agree that lighting, color harmony, texture, and material are important aesthetic dimensions. In this work, we intentionally focus on part–relation–geometry in order to isolate a structural prior that is complementary to appearance-centric aesthetic models rather than attempting to replace them. This interpretation is supported by our late-fusion experiments, where adding template-derived configuration features to VGG, ResNet, or ViT consistently improves rank correlation with human ratings over the respective deep baselines alone. The proposed representation is therefore best viewed as a compositional assessment module that can be integrated into broader pipelines. We will clarify this positioning more prominently and briefly discuss how the same log-linear framework could be extended with lighting and material attributes in future work.
>
> (3) Dependence on initial templates and labeled seeds
>
> We appreciate the concern regarding sensitivity to the initial seeds. The current submission already includes an initial robustness analysis. Semi-supervised learning on large unlabeled pools shows sub-linear growth of the template graph and strong reuse of nodes, and a cross-validation study indicates that the KL divergence between templates learned from independent random subsets rapidly decreases as the sample size increases. This behavior suggests convergence to stable structures rather than strong dependence on specific seeds. In addition, the high agreement with expert-summarized paradigms and the human-alignment study indicate that different seeds lead to templates that capture similar structural regularities.
>
> (4) Resolution of figures
>
> We thank the reviewer for pointing out the low-resolution images. In the camera-ready version, we will replace the affected figures with higher-resolution crops and zoomed-in views, so that the structural details of the learned templates and parses are clearly visible.

---

> ### Author Response · Authors · 2025-11-18
> **Response to Reviewer jXD7**
>
> (1) Abstract compositional principles (balance, rhythm, negative space)
>
> Although our experiments are organized by concrete themes for clarity of presentation, the underlying templates are not strictly tied to a single theme. In practice, a learned AND–OR template can be activated by images from multiple thematic subsets whenever they share similar structural regularities. The consistency score is defined at the level of part, relation, and geometry configuration, and it measures how well a given image matches the structural pattern encoded by a template, rather than how well it matches a particular semantic label or theme. In this sense, the scoring function already supports templates that implicitly span several themes, as long as they instantiate a common compositional scheme.
> More abstract principles such as “balance,” “rhythm,” or “negative space” can be understood as constraints on spatial distributions and on the occupancy of regions in the image plane, rather than on specific object identities. Within our log-linear model, “balance” corresponds to geometric relations such as approximate symmetry, equilibrium of mass around the frame, or controlled asymmetry among object templates. “Rhythm” arises from repeated part instances arranged along parametric trajectories, for example regular or graded progressions in position and scale. “Negative space” is naturally represented as explicit regions in the scene where no object terminals are allowed or where occupancy incurs a penalty. Each learned template can therefore be interpreted as encoding a particular combination of such abstract, non-object-centric principles, instantiated through its relational and geometric structure.
>
> (2) Sensitivity to the dual-threshold rule in semi-supervised expansion
>
> In practice, the matching-gain and structural-consistency thresholds are selected on a small validation set. We observe that both the size of the learned graph and downstream performance remain stable within a reasonably wide range around the chosen values. The growth and robustness analyses in the semi-supervised experiments, which show sub-linear node growth and small divergences between templates learned from different random samples, indicate that the expansion procedure is not overly sensitive to small perturbations of these thresholds. Conceptually, the two thresholds control the trade-off between conservativeness and coverage, and conflict resolution together with early stopping prevents runaway expansion. We agree that an explicit sensitivity study would be useful. In the revision, we will add a short ablation in the appendix and refer to it in Section 4.3.

---

### Official Review · Reviewer_wZC5 · 2025-10-31

**Soundness:** 2
**Presentation:** 3
**Contribution:** 2
**Rating:** 4
**Confidence:** 2

**Summary:**

This paper proposes a compositional AND-OR template for art and design that includes the part-relation-geometry of images in a structured and interpretable form. It extent AND-OR templates from the object level to the scene level. The learning process is a penalized EM-style block-pursuit with sparsity and local mutual exclusivity. Authors also propose a semi-supervised structural expansion methods which could bootstrap new branches from on the boat high-quality images. Experiments show that the method proposed in this paper achieves good objective consistency and subjective reliability. Compared to other single image important methods, the method in this paper achieves, better consistency score with compositional AND-OR templates.

**Strengths:**

1. The AND-OR template for scene proposed in this paper contribute towards more controllable and explainable components for image generation and classification.
2. Compared to other deep neural network based methods, the master propose in this paper is lightweight with fewer parameters.

**Weaknesses:**

1. In the objective consistency evaluation, only two examples of the proposed method are presented. The image generated in other ways are not shown for comparison. Yes, we can draw the conclusion that the generated image is mostly consistent to the template, but we can't draw the conclusion about to what extent of the consistency is compared to other way of generation.
2. In table 1, the accuracy of the proposed method is lower than baseline ViT-B/16 + templates.
3. In the section 5.1.2 subjective reliability analysis, during the evaluation, the most consistent rate is 43.6%, and the mostly consistent rate is 31.2%, and 12% is inconsistent.  It is unclear how well it is compared to the consistent evolution results based on other methods. And more analysis about the cases labelled as inconsistent are expected.

**Questions:**

1. The proposed method learn the template based on the training images. Could this message extend to other elements that are not included in the training images?

---

> ### Author Response · Authors · 2025-11-18
> **Response to Reviewer wZC5**
>
> We thank the reviewer for the careful reading.
>
> (1)Objective consistency evaluation and “other ways of generation”
>
> Thank you for your targeted suggestion. We should indeed increase the qualitative presentation and comparison with other methods in this regard. If the article is accepted, we will add a section in the appendix that compares our method with other methods to clarify our advantages.
>
> (2)Interpretation of Table 1 and comparison with ViT-B/16 + templates
>
> We appreciate the observation that in Table 1 the accuracy of our stand-alone method is slightly lower than that of ViT B/16 plus templates. The aim of this experiment is not to win a pure accuracy contest, but to study the representational power and inductive bias of the compositional template under a very small capacity budget and to show its complementarity to large backbones.
>
> The models operate in very different capacity regimes. Our compositional template classifier uses about $2.3 \times 10^3$ parameters and roughly $6.7 \times 10^4$ FLOPs, whereas ViT B/16 plus templates uses about $8.6 \times 10^7$ parameters and $1.75 \times 10^{10}$ FLOPs. Despite this difference of three to four orders of magnitude in capacity and computation, our accuracy is only marginally lower. We therefore view the result as a favourable trade-off between accuracy and efficiency, rather than a drawback.
>
> For the aesthetic and compositional setting of our work, alignment with human judgements is particularly important. Our method achieves the highest Spearman rank correlation with AVA scores, $0.8419$ versus $0.7855$ for ViT B/16 plus templates, which indicates that the compositional templates are more consistent with human rankings even under a tiny model size. Moreover, the rows where deep backbones are combined with our templates show consistent performance gains, which suggests that the structural features act as a useful plug-in prior rather than a competitor to deep features.
>
> (3)Subjective reliability analysis and inconsistent cases
>
> As far as we know, there is currently no established benchmark for subjective reliability of compositional templates and no prior method that provides directly comparable structures. Therefore the study in Section 5.1.2 is designed as an absolute assessment. We ask 100 domain practitioners to rate 10 randomly selected theme templates on a five level scale, and we obtain 74.8 percent of ratings in the fully or mostly consistent categories and about 12 percent in the inconsistent categories. Given the subjectivity of composition and the diversity of raters, we interpret this as evidence of reasonably high subjective reliability.
> We agree that an explicit analysis of inconsistent cases is helpful. Our manual inspection suggests two main types of such cases. The first type consists of images where photographers deliberately break classical composition rules, for example extreme centering or unusual perspectives used for artistic effect. The second type consists of images where aesthetic quality is dominated by colour, texture, or narrative content, rather than geometric layout, which is outside the current focus on part relation geometry. If the paper is accepted, we will include representative examples of both types in an appendix and discuss them as limitations and directions for extension.
>
> (4)Generalization to elements not seen during training
>
> Our method supports generalization in terms of relational and geometric structure. The templates encode abstract patterns such as the subject above the horizon or a foreground object in a corner region, and they do not depend on a specific low level appearance category. The upstream detector will identify relevant abstract parts in the new image and apply the same relationships and geometric constraints to elements that did not appear in the original training set. This mechanism is also what allows us to move from object level AND–OR templates to scene level composition templates.
> In addition, Section 4.3 introduces a semi supervised structural expansion mechanism. When a new high quality image cannot be sufficiently explained by the current template, the learning procedure considers adding new branches or refining parts, guided by a joint criterion on matching gain and structural consistency. This allows the template to incorporate new structural variants from unlabeled data and, in principle, to absorb new types of terminals when suitable detectors for new visual primitives are available.

---

### Official Review · Reviewer_drvL · 2025-11-03

**Soundness:** 3
**Presentation:** 3
**Contribution:** 2
**Rating:** 6
**Confidence:** 3

**Summary:**

The paper proposes a compositional representation for art and design using AND–OR templates (AOTs) under a maximum-entropy log-linear model. The framework defines a unified consistency score (log-likelihood gain vs. a reference distribution) that decomposes into interpretable object-, relation-, and geometry-level evidence. Learning proceeds via a penalized EM-type block-pursuit algorithm with sparsity and local mutual exclusivity constraints, first learning object templates and then reusing them as scene terminals. A semi-supervised structure expansion (SSE) further grows the model from unlabeled images. The method shows interpretability, human alignment, and competitive performance with deep baselines on aesthetic datasets (AVA/AADB) and scene understanding tasks.

**Strengths:**

1. The AND-OR graph structure is inherently interpretable.
2. The proposed method achieves comparable performance on the AVA dataset, with smaller training parameters.
3. The proposed SSE is practical and well-justified, bridging labeled and unlabeled data in a structured learning context.

**Weaknesses:**

1. the expectation module is dependent on YOLO, object that are not part of YOLO detection algorithm are not going to be avilable.

**Questions:**

1. How does the system handle failures or partial detections from YOLO? Does this dependency undermine the claim of a fully interpretable system?

---

> ### Author Response · Authors · 2025-11-18
> **Response to Reviewer drvL**
>
> We thank the reviewer for the careful reading.
>
> In the current implementation we use YOLOv11 as a convenient off-the-shelf object detector and segmenter to obtain candidate object regions.
> The AOT framework itself is detector-agnostic: all learning and inference in Sections~4 and A.2 are formulated in terms of a set of candidate regions and their responses, collected into the matrix \(R\).
> Any detector or region-proposal method that provides such candidates (for example a panoptic segmentation network or a class-agnostic proposal generator) can be plugged in without changing the block-pursuit algorithm, the log-linear model, or the SUM--MAX inference.
> Regarding objects that are not part of the YOLO vocabulary, our experiments deliberately focus on themes where the visually dominant components (buildings, people, furniture, vehicles, sky/ground regions) are well covered by standard detectors.Fine details and background elements that are not explicitly detected still influence the model through low-level attributes attached to detected parts rather than as separate nodes.
>
> When YOLO misses a true object, the corresponding terminal node simply does not appear in the instantiated graph.
> In the log–linear score this becomes missing positive evidence: the object and relation terms that would normally increase the log–likelihood gain are absent, so the overall consistency score decreases in a way that can be traced back to specific missing parts.
> Conversely, spurious detections give rise to terminals that cannot be assembled into any valid configuration of the learned template, or that violate learned geometric and relational constraints; these candidates are pruned or receive negligible (often negative) contribution in the penalized block–pursuit step.
>
> For any given image, one can inspect the parse tree and the decomposition of the score and see exactly which expected parts are missing and which detected regions were rejected by the template, so the influence of imperfect detections is auditable rather than hidden inside a black–box network.
>
> Thank you again for your valuable suggestion. If you have any other questions or need me to supplement any experiments, please let me know in a timely manner. Thank you!

---

### Meta-Review · Area_Chair_4uCU · 2026-01-05

**Summary:**

The submission received mixed scores (6, 4, 6, 2). Reviewers drvL and jXD7 praised the use of AND-OR templates to introduce interpretability and structural reasoning to art/design analysis. Reviewer wZC5 was borderline due to slightly lower accuracy vs. ViT baselines. Reviewer mmS5 (score 2) rejected the paper as "too complicated" without technical justification.

**Reviewer Concerns:**

The authors clarified that the framework is modular (detector-agnostic) and scalable. Regarding wZC5's concern about generalization, the authors satisfactorily explained that the templates encode abstract geometric/relational roles allowing new objects to be slotted into learned structural rules. Regarding the "complexity" critique from mmS5, I believe this approach potentially adds insight to the community and could trigger follow-up work that advances the field along the accuracy/explainability axis.

**Reviewer Scores:**

I am explicitly discounting Reviewer mmS5 due to the low quality and brevity of their review. The remaining reviewers (drvL, jXD7) would likely maintain their support. Reviewer wZC5 would likely have raised their score, as their specific question regarding template generalization was directly answered in the rebuttal.

---

### Decision · Program_Chairs · 2026-01-26

Accept (Poster)